# Sub-$k_BT$ micro-electromechanical irreversible logic gate

M. López-Suárez[1], I. Neri[1,2] & L. Gammaitoni[1]

In modern computers, computation is performed by assembling together sets of logic gates. Popular gates like AND, OR and XOR, processing two logic inputs and yielding one logic output, are often addressed as irreversible logic gates, where the sole knowledge of the output logic value is not sufficient to infer the logic value of the two inputs. Such gates are usually believed to be bounded to dissipate a finite minimum amount of energy determined by the input–output information difference. Here we show that this is not necessarily the case, by presenting an experiment where a OR logic gate, realized with a micro-electromechanical cantilever, is operated with energy well below the expected limit, provided the operation is slow enough and frictional phenomena are properly addressed.

[1] NiPS Laboratory, Dipartimento di Fisica e Geologia, Università degli Studi di Perugia—via Pascoli, I-06123 Perugia, Italy. [2] INFN Sezione di Perugia—via Pascoli, I-06123 Perugia, Italy. Correspondence and requests for materials should be addressed to M.L.S. (email: miquel.lopez@nipslab.org) or to I.N. (email: igor.neri@nipslab.org).

In recent years, there has been an exponential growth in microprocessor computing capability, due to the increased ability to put a larger (and, still, increasing) number of transistors inside the same chip volume. The increase in the number of transistors has been accompanied by a corresponding increase in the amount of produced heat, despite the marked drop in the amount of energy dissipated per switch operation[1]. Such a large heat production is, presently, considered a potential road-block for future scaling over the next 10–15 years[2]. Innovative solutions to markedly decrease the dissipated heat are presently being researched in many laboratories through the world. This, together with a significant growth of experimental capabilities in energy measurements in nanoscale systems[3–6], has fuelled a resurgence of interest in the so-called thermodynamics of information[7].

In present day automatic computing, logic gates are the building blocks of many ICT (Information and Communication Technology) devices. Here, the computation is carried out by networking logic gates to perform all the required logical and arithmetical operations. A single logic gate is made by interconnecting one or more electronic transistors employed as logic switches, as in the example depicted in Fig. 1a. A logic switch is a device that can assume physically distinct states, as a result of external inputs. Usually the output of a physical system assumes a continuous value (for example, a voltage), and a threshold is used to separate the output physical space into two or more logic states. If there are two states (we can call them S0 and S1), we have binary logic switches. Devices realized with logic switches can be divided into two classes underpinned by combinational or sequential logic circuits. Combinational logic circuits are characterized by the following behaviour: in the absence of any external force, under equilibrium conditions, they are in the state S0. When an external force $F_0$ is applied, they switch to the state S1 and remain in that state as long as the force is present. Once the force is removed they go back to the state S0. Examples are electromechanical relays and transistors. This is the case of a logic gate built with mechanical switches, as represented in Fig. 1b. Sequential logic circuits are characterized by the following behaviour: if they are in the state S0, they can be changed into the state S1 by applying an external force $F_{01}$. Once they are in the state S1, they remain in this state even when the force is removed. The transition from state S1 to S0 is obtained by applying a new force $F_{10}$. Once the logic circuit is in the state S0, it remains in this state even when the force is removed. In contrast to the combinational logic circuit, the sequential one remembers its state even after the removal of the force. This memory lasts for a time that is short compared with the system relaxation time. In fact, if one waits long enough, the sequential switch relaxes to equilibrium that, in a symmetric device, is characterized by a 50% probability to be in the S0 state and 50% probability to be in the S1 state. In all practical cases the relaxation time is usually much longer than any operational time; hence the sequential logic circuit can be considered a system that remembers the last transition produced by the application of the short duration external force. Examples include electronic flip-flop and the complex 'storage capacitor + transistor' used in present DRAM (dynamic random access memory). They are employed in computers to perform the role of registers and memory cells.

In this paper, we report the results of energy dissipation measurements on a practical realization of a combinational logic circuit that implements the OR logic gate, realized with a micro-electromechanical cantilever activated by electrostatic forces, showing this can be operated well below $k_BT$.

## Results

**Device description and experimental set-up.** The device consists of a single logic switch made with a $Si_3N_4$ cantilever that can be bent by applying electrostatic forces with two electrical probes close to the cantilever tip (Fig. 2a). Experimental set-up details are presented in the Methods section and are depicted in Supplementary Fig. 1. The set-up calibration is presented in Supplementary Methods: experimental set-up calibration. The logic state of this device is encoded in the tip position as

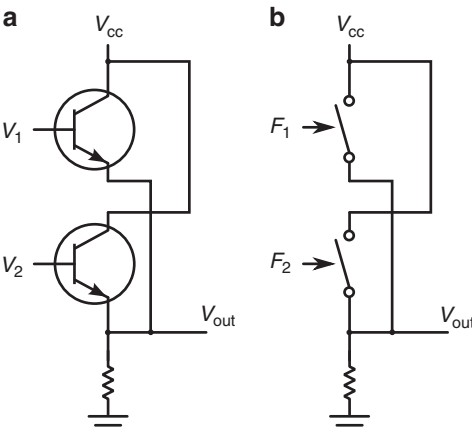

**Figure 1 | OR logic gate.** (**a**) An OR logic gate realized interconnecting two transistors. The output of the logic gate depends on the combination of inputs encoded by voltages $V_1$ and $V_2$. The state of the logic gate returns to its original state once the inputs are removed. (**b**) An OR logic gate realized interconnecting mechanical switches.

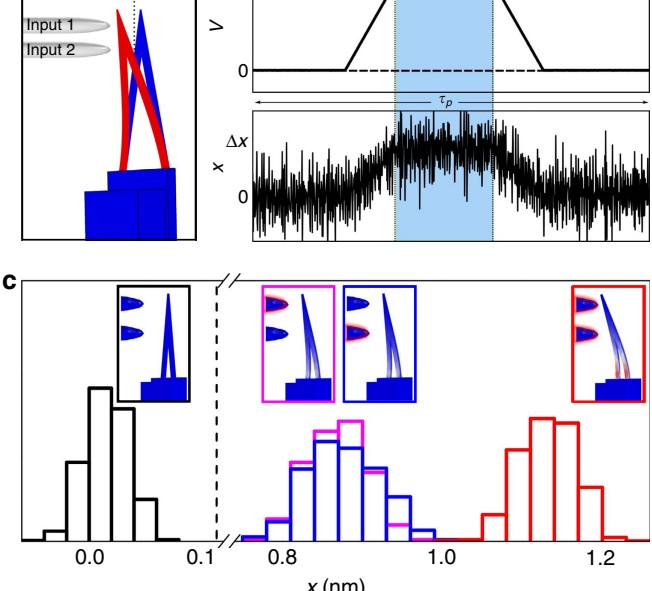

**Figure 2 | Schematic of the logic gate operation mode.** (**a**) Inputs are the forces acting on the cantilever through the electrodes. The position of the cantilever tip encodes the output of the logic gate. (**b**) Value of the electrode voltage $V$ (logic input) and the consequent displacement of the cantilever tip $x$ (gate output). (**c**) Statistical distribution of the cantilever tip position as a function of the four possible inputs (that is, 00, 01, 10 and 11); the threshold value for the OR gate is represented by the dashed line. By changing the position of the dashed line, the gate can be operated also as an AND gate.

depicted in Fig. 2a. The input of the logic gate (I1, I2) is associated with the voltage of the respective electrical probe. As mentioned before, our logic gate is operated in direct analogy with transistor-based logic gates: when the bias voltage input is applied to the transistor gate, the channel is opened and the conducting state of the transistor changes the voltage of the output terminal. In our cantilever, once the electrostatic force is applied, the position of the cantilever is changed. In the transistor, once the bias voltage input is zeroed, the channel closes and the initial state is recovered. The same happens in our cantilever: once the force is removed, the cantilever position goes back to the zero-force state. Clearly, both devices belong to the combinational device class and both can be employed to make logic gates. One relevant difference between the device considered in this work and commercial logic gates is that, in our case, the input and output have different natures (electrostatic forces and mechanical displacement respectively), while for the transistor systems both are voltages.

In this scenario, we can operate the cantilever as an OR gate by carefully setting the threshold for the two states S0 and S1. In Fig. 2c, the statistical distribution of the position of the cantilever tip is reported as a function of the inputs. We observe that logic states S0 and S1 are clearly distinguished, since the overlap between the corresponding probability density functions is negligible. Moreover, logic inputs corresponding to the mixed states (I1 = 1, I2 = 0 and I1 = 0, I2 = 1) produce a physically indistinguishable position distribution and thus the same logic state S1. Finally, the logic input I1 = 1, I2 = 1 produces a larger displacement that, due to the threshold setting, belongs to the same logic partition S1. Under these circumstances the cantilever-based gate performs like an OR gate that is a logically irreversible device: in fact there is at least one case where, from the sole knowledge of the logic (and the physical) output, it is not possible to infer the status of the logical inputs. Having established this point, we address the issue of the minimum energy required for operating this logic gate.

**Work and heat production estimation**. Our measurement strategy is the following: we plan to operate the logic gate by switching from the S0 state to the S1 state and from S1 state to the S0 state. We measure the work $W$ performed on the system by the external forces during this entire operation cycle. Since, in a cycle the change in the internal energy $H$ of the system is zero, we have $Q = W$. To measure the work performed during the gate operation, $W$, we compute the Stratonovich integral as in[8,9]

$$W = \int_0^{\tau_p} \frac{\partial H(x, V)}{\partial V} \dot{V} \mathrm{d}t \qquad (1)$$

where $H(x, V)$ is the total energy of the system, $x$ the tip displacement, $V$ the input voltage and $\dot{V}$ its time derivative. The integral is computed during a single operation cycle of duration $\tau_p$. During this cycle (Fig. 2b) if the $i$th logical input I$i$ ($i = 1, 2$) is set from 0 to 1, the voltage of the $i$th probe increases linearly from $V = 0$ to $V = V_0$ during a time interval of length $\tau_p/8$. The same time interval $\tau_p/8$ occurs when decreasing the voltage from $V = V_0$ to $V = 0$ to set the input from the logic 1 to 0. In this experiment, we start every operation cycle with all inputs set to 0; hence, no initial force is applied to the cantilever. According to this, the logic output is read after a time $\tau_p/8$, for a time interval of $\tau_p/4$; see the blue highlight in Fig. 2b. We would like to emphasize that the output of our logic gate can be fed directly as an input to a second similar device (as shown below) and so on, to perform all the calculations desired. Clearly, our logic gate is a combinational device; thus, the removal of the inputs makes the system revert to the initial state S0 (regardless of whether the final

state was S0 or S1). If we want to remember the final state, we need to couple this device to a sequential device where a Landauer reset[10] might be required and a minimum dissipation of $k_B T \log 2$ is needed.

The protocol duration $\tau_p$ and maximum voltage $V_0$ are the two control parameters that set the speed of the operation and the maximum displacement, respectively (expressions for the forces are reported in Supplementary Methods: force calibration). To measure the energy dissipation, the logic gate is operated addressing all the possible input combinations. Each combination is applied for different protocol duration $\tau_p$. For each combination of $\tau_p$ and $V_0$, $Q$ is evaluated over $\sim 1,000$ repetitions.

In Fig. 3a, we show the produced heat $Q$ in units of $k_B T$ as a function of the protocol duration for three different sets of logic inputs: '01', '10' and '11'. For this case $V_0$ is set to 2.5 V. As we mentioned above, the measured heat is a random quantity whose statistical distribution is well reproduced by a Gaussian curve (Fig. 3b). It is interesting to note that, although the average value of the dissipated heat is positive, the distribution shows also a negative tail. According to the experimental results presented in Fig. 3a, we can conclude that the dissipated heat can be reduced well below $k_B T$ if the protocol duration is extended in time.

The claimed linkage[11] between logical and physical irreversibility has animated a long debate[12]. Although recent studies[13] have contributed to clarify this aspect from a purely theoretical point of view, it remains widely controversial and is still missing experimental verification. Our experiment rules out the presence of a finite 'minimum dissipated heat' due to logical irreversibility, an argument often invoked when the reduction of input–output information is considered. We stress here that our experiment does not question the so-called Landauer-reset interpretation, where a net decrease of physical entropy requires a minimum energy expenditure[10,14]. What we have here is a logically irreversible computation, that is a generic process where a decrease in the amount of information between the output and the input is realized with an arbitrarily small energy dissipation; this shows that logical reversibility and physical reversibility have to be treated on independent bases[12].

**Dissipative mechanisms**. In the following, we focus our attention on the dissipative mechanisms that are activated during the cantilever operation. According to Fig. 3c, the average dissipated heat is proportional to the square of the displacement amplitude and follows a protocol duration power law. This can be explained introducing a dissipation model that considers the coexistence of different frictional mechanisms. According to the Zener theory[15], the dissipated energy can be computed as the work done by the frictional forces expressed in terms of the loss angle $\phi$ (see the Methods section for details)[16–19]. Thus, the total dissipated heat during the frictional dynamics is proportional to $\phi$ and to the square of the bending amplitude $\Delta x^2$. The dependence on the time protocol $\tau_p$ follows from the frequency dependence of $\phi(v) = \phi(1/\tau_p)$ and thus it decreases in time according to a power law having components in $\tau_p^{-3}$, $\tau_p^{-1}$ and a constant term, in good agreement with the experimental data in Fig. 3c.

**NOR logic gate**. On the basis of the results just presented we argue that, in principle, we can design and operate a 'towards zero-power' computer[14] based on micro-electro-mechanical system (MEMS) cantilevers. In fact the position of one cantilever can influence the position of a second one if they are properly polarized. As an example in Fig. 4, we present the results for a NOR gate by coupling the OR gate that we considered earlier with an additional electrically polarized cantilever (Fig. 4a). Since, the two cantilevers are biased with different

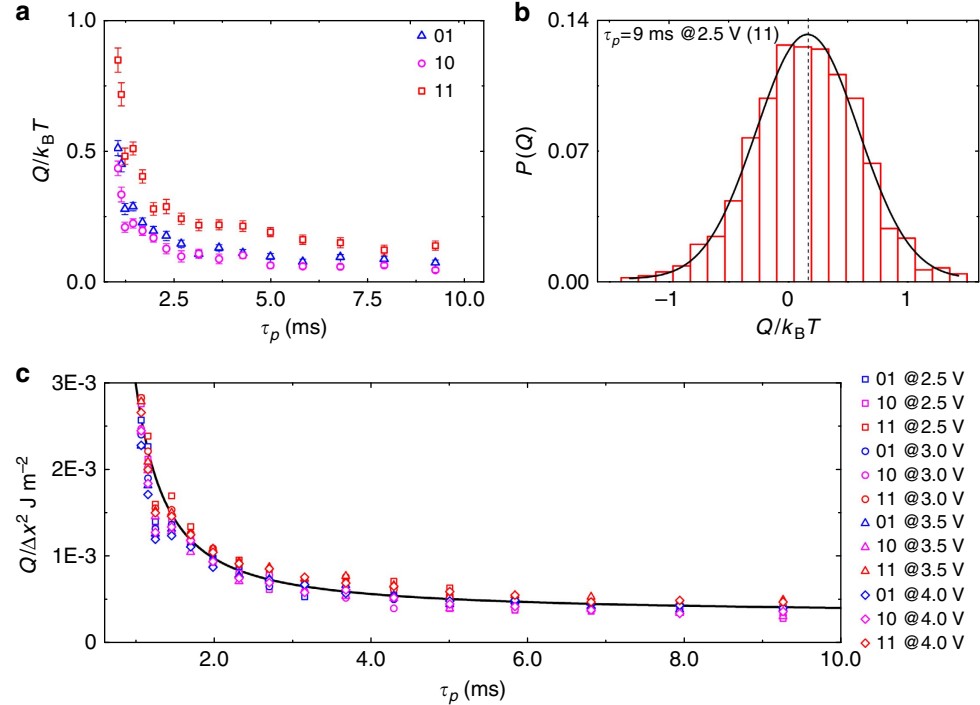

**Figure 3 | Heat production during operation of the logic gate.** (**a**) Average produced heat as function of protocol time $\tau_p$, for the three different sets of inputs (that is, '01', '10' and '11'). With increasing protocol time the produced heat decreases following a power law. (**b**) Heat distribution $P(Q)$ for the case $\tau_p = 9$ ms, input '11'. Red bars represent the histogram of the measured distribution, black line represents the fit with a Gaussian distribution, and black dashed line the average value of produced heat. It is interesting to note that, although $<Q> > 0$, the distribution of the generated heat has negative tails. (**c**) Mean produced heat normalized over the square of maximum displacement of the cantilever tip as a function of the protocol time $\tau_p$. The produced heat is evaluated for all combinations of the input for different values of voltages applied to the input probes. For equal protocol time, $\tau_p$, all points show approximately the same value in agreement with the dissipation model presented. Solid line represents the fit with the dissipation model.

voltage they interact electrostatically. If the gap between the two cantilevers is set properly it can act as a NOR gate that, being a universal logic gate, can be used to perform any general logic operation. In Fig. 4c, we present an experimental time series of the tip position of the original OR gate, as function of the input configurations, together with the tip position of the added cantilever, that is, NOR. We have evaluated the dissipated energy associated with the transfer of the output signal of one cantilever to the input of the next one. In the present NOR case, we have measured the displacement of the two cantilevers in the adiabatic condition allowing to evaluate the heat production by means of the Zener model presented above. In particular, the displacement of both cantilevers relative to the four combination of input has been measured. Assuming to work in the adiabatic condition, that is, large $\tau_p$, the average heat $<Q>$ has been estimated from the relation obtained by the fit presented in Fig. 3c, combining the measured displacements and the relation $Q/\Delta x^2$. In the adiabatic condition, we can consider only the structural damping, since the others contributions goes to zero increasing the protocol time. An average energy dissipated of $<Q> = 0.31 \pm 0.05\ k_B T$ is then obtained.

**Full-adder**. To prove that we can operate complex logic operations, we have conceived a single bit full-adder. The output of the full-adder is composed by two bits, the sum S and the carry output $C_{out}$. The latter can be simply computed with a single cantilever fed with the three inputs, by setting properly the logic threshold. Computing the sum bit S requires a more complex device composed by a set of four cantilevers, where the logic output is encoded in the position of the last one. The schematic of

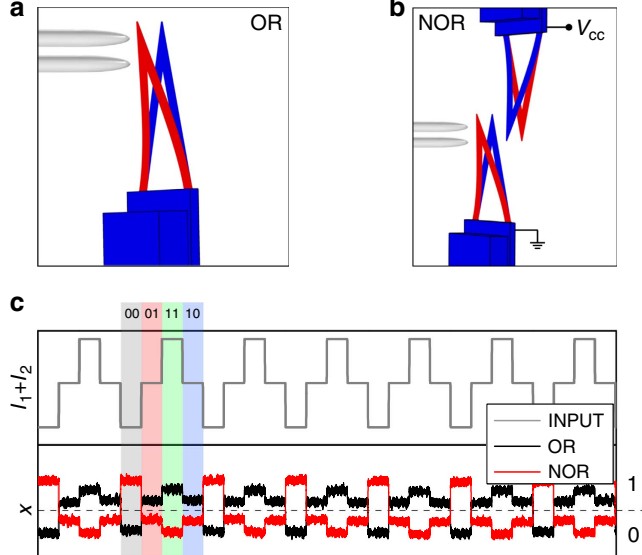

**Figure 4 | Complex operations coupling more mechanical gates.** The single OR gate (**a**) can be negated producing a NOR gate (**b**) by means of an additional electrically polarized cantilever, placed in front of the first one. (**c**) Outputs of the two configurations are reported as function of the possible inputs, highlighted in grey (00), red (01), green (11) and blue (10). Setting a threshold, represented by the dashed line, it is evident that the two configurations act as an OR or NOR gate.

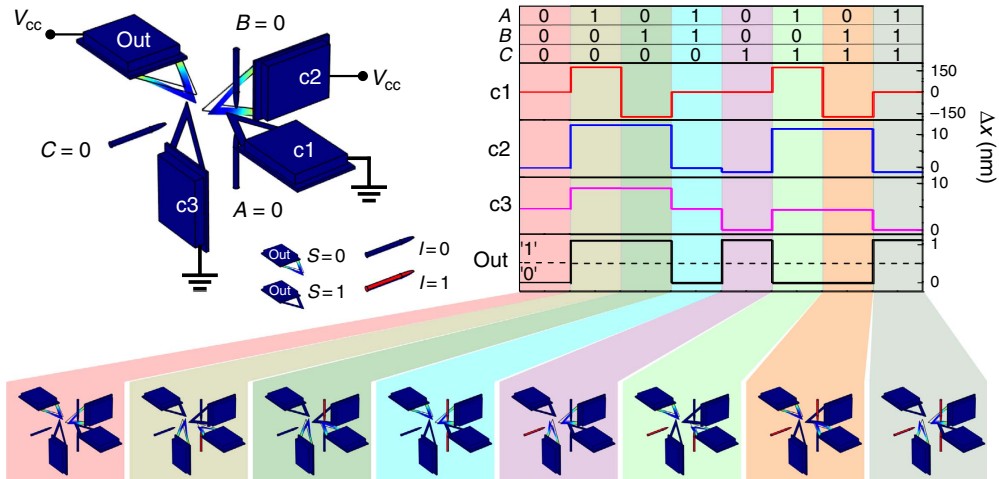

**Figure 5 | Full-adder sum bit.** Top: full-adder sum calculation realized with four cantilevers coupled with electrostatic forces. The deflection of one cantilever depends on the inputs and the position of the nearest cantilevers. Bottom: deflection of each cantilever as function of all combination of inputs (first row of the table). The last cantilever encodes the output sum bit of the full-adder.

the conceived structure is presented in Fig. 5. We simulated numerically the dynamics of the coupled system and the results are presented in Fig. 5 for all combinations of inputs. As it is apparent the last cantilever (out) encodes correctly the logic output of the sum bit S of the full-adder. The bottom part of Fig. 5 shows the final deflection of the cantilevers for the relative inputs. The full-adder can be operated reversibly by controlling precisely the velocity of each cantilever.

## Discussion

In conclusion, we have presented an experiment where we showed that a combinational logic circuit that implements the OR gate, realized with a micro-electromechanical cantilever, can be operated with energy well below $k_BT$, at room temperature, if the operation is performed slowly enough and friction losses are minimized. Thus, no fundamental energy limit need be associated with irreversible logic computation in general and physical irreversibility is not necessarily implied. Moreover, we have shown that by directly coupling two electromechanical cantilevers, we can realize and operate a universal NOR gate with energy dissipation below $k_BT$. This implies that complex-networked structures can be realized by direct coupling without adding significantly to dissipation. As an example, we have designed and simulated the operation of the sum bit of a full-adder.

## Methods

**Experimental set-up details.** All measurements are carried on in a vacuum chamber at $P = 10^{-3} \pm 0.01$ mbar and at room temperature ($T = 300$ K). The mechanical structure is a 200-μm long V-shaped cantilever providing a nominal stiffness, $k = 0.08$ N m$^{-1}$, with a first-mode resonant frequency of $f_r = 14,950 \pm 1$ Hz and a quality factor $Q = 2,886 \pm 10$, resulting in a relaxation time $\tau = 61.4 \pm 0.2$ ms. The deflection of the cantilever, $x$, is measured by an atomic force microscopy-like optical lever: the deflection of the laser beam (633 nm) due to the bend of the cantilever is detected by a two quadrants photo detector. The response of the photo detector in the linear regime is $x = r_x \Delta V_{PD}$ with $r_x = 2.1256 \cdot 10^{-8}$ m V$^{-1}$. Position and voltage measurements were digitalized at 50 kHz.

**System energetics.** The total energy of the system is expressed as $H(x,V) = H_{kinetic} + H_{int}(x) + H_{ext}(x,V)$. Assuming the linear response of the structure for small displacements, the internal potential energy is $H_{int}(x) = \frac{1}{2}kx^2$. The protocol time duration considered in the experiment is shorter than the relaxation time, $\tau_P < \tau$, and thus at the end of the protocol part of the energy is stored as kinetic energy. However, the assumption of $\Delta H = 0$ is still valid because,

on average, the initial and final kinetic energies are the same. Input forces are applied through two electrostatic probes consisting of two tungsten tips (100-nm tip radius) placed at $g = 5$ μm. The resulting electrostatic force can be approximated by $F = \alpha \frac{V^\gamma}{(g-x)^2}$ where $\alpha$ and $\gamma$ depend on the input ('01', '10' or '11'), thus $H_{ext}(x,V) = -\alpha \frac{V^\gamma}{(g-x)}$.

**Dissipative model based on Zener theory.** The dissipative model behind the power law fit in Fig. 3c is obtained by the Zener theory, assuming that the dissipative dynamics can be expressed as the result of frictional forces that represent the imaginary component of a complex elastic force $-k(1+i\phi)$ (ref. 15). In general, $\phi$ is a function of frequency and for small damping it can be expressed as the sum over all the dissipative contributions. In our case $\phi(v) = \phi_{str} + \phi_{th-el} + \phi_{vis} + \phi_{clamp}$. Here $\phi_{str}$ is the structural damping[16] ($\phi$ is independent of the frequency $v$), $\phi_{th-el}$ and $\phi_{vis}$ are the thermo-elastic[17,18] and viscous damping that can be assumed to be proportional to the frequency for frequencies much smaller than the cantilever characteristic frequency, and $\phi_{clamp}$ represents the clamp recoil losses ($\phi(v) \propto v^3$) (ref. 19).

**Data availability.** The data that support the findings of this study are available from the corresponding authors upon request.

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

## Acknowledgements

We gratefully acknowledge financial support from the European Commission (FPVII, grant agreement no: 318287, LANDAUER and grant agreement no: 611004, ICT-Energy), Fondazione Cassa di Risparmio di Perugia (Bando a tema Ricerca di Base 2013, Caratterizzazione e micro-caratterizzazione di circuiti MEMS per generazione di energia pulita) and ONRG grant N00014-11-1-0695. We thank D. Chiuchiù, C. Diamantini, C. Trugenberger, G. Carlotti, M. Madami, F. Marchesoni and A. R. Bulsara for useful discussions.

## Author contributions

M.L.S. and I.N. designed the experiment, performed the measurements and analysed the measured data. L.G. supervised the data analysis and provided the dissipative model. All authors contributed to the writing of the manuscript.

## Additional information

**Competing financial interests:** The authors declare no competing financial interests.

