## [Peer Review File · Nature Communications]

Transferred manuscripts:

Reviewer #1 (Remarks to the Author):

I believe the authors have seriously considered my comments and made appropriate changes. I believe now the paper is ready for publication in Nature Communications as it is timely and important in the field thermodynamics; as I described in my previous report.

One minor point: In the introduction the authors write "This lack of linkage between logical and physical irreversibility". If I understand this correctly, the sentence should allude to a preceding discussion of the difference between logical and physical irreversibility. But as the words "physical irreversibility" are not mentioned until this point, the sentence makes little to no sense. I like the sentiment, but as written it is problematic.

Reviewer #2 (Remarks to the Author):

I would like to thank the authors to revise the manuscript following the reviewer's comments. The manuscript was improved and it is easier to understand the concept that authors argue. I now understand that the point raised by the authors is that the total "dissipated" energy per cycle within the mechanical system can be smaller than kT . It was my fault that I did not fully understand this point. However, it is now rather clearer for me what is missing in the discussion and I believe that the manuscript does not include enough importance to warrant the publication in Nature Communications.

The energy cost for calculation should be evaluated between the initial configuration and the final configuration including all the related physical systems. In the final configuration, the calculation result should be readable otherwise the result cannot be utilized after the calculation. Therefore the information should be somewhere stored even in their-called "combinational" logic. Also in the present transistor-based logic system, the final results should be extracted from outside to get the results of calculation. Generally the dissipated energy of transistor-based devices might be evaluated only within the focused device itself but I believe that this is because the energy is much larger than kT and the energy used in the followed device to receive the transferred binary information can be neglected.

My largest question is; what is the reasonable definition of the energy required for computation. It is of course possible to define it as the energy spent only within the focused "combinational" device itself as authors did in this manuscript. However, if it is the case, the argument that "the operation energy can be smaller than kT " is confusing and misleading. The calculation result can be utilized only when the information was transferred to the next device and the energy consumed there should be discussed as the part of a whole calculation process. In the present setup, the position of cantilever read by the laser system changes some physical state there and an additional energy larger than $kT \log_2$ will be dissipated when the information is erased. I noticed that the authors already recognized this point as described in page 6 of revised manuscript but their discussion is still based on the energy dissipated only in the mechanical system.

The second question is on the sequential operation, raised from the application viewpoint, i.e. the

fan-out. The energy dissipation might be lower than kT within the single device, but I am not sure if the output is readable in the followed similar device. The optical method has enough high position sensitivity to distinguish the two states but transferring the information to next cantilever requires similarly high position sensitivity. I believe that some buffer device is required to amplify the signal and the energy spent therein should be considered.

In summary, I now agree with the argument given by the authors in this manuscript that the "dissipated" energy per cycle within the focused mechanical device can be smaller than kT . However, I believe that the experiment is straightforward and the statement, "the device can be operated with energy lower than kT " is true but highly misleading because the energy dissipated in the input and output system, as well as the amplifier required for the sequential operation was not taken into account. Therefore, I cannot recommend the publication of this manuscript. A nice demonstration was given by for example ref. 8, where the energy required to erase the stored information was also taken into account in order to prove the asymptotic approach to the Landauer's limit.

Reviewer #3 (Remarks to the Author):

I have read the paper "Sub $k_B T$ micro electromechanical irreversible logic gate" by Lopez-Suarez, Neri and Gammaitoni. The paper deals with a very interesting topic, the information thermodynamics, which attracts much attention. Recently, several experiments relevant for this topic have appeared as cited in the paper [Refs. 6,7,8,9]. In the paper, the authors stressed the difference between the logical irreversibility and the thermodynamic irreversibility. This is an important issue and has been theoretically discussed for a long time. Recently it has been a major interest in the community because of advances in work and heat measurement techniques of small systems.

The previous experiments [Refs. 6,7,8,9] have been more or less related to the Landauer's erasure principle. The memory erasure process is logically irreversible, but can be thermodynamically reversible. Although the minimum energy cost $k_B \log 2$ has been verified experimentally [Refs. 8 and 9], thermodynamic reversibility of the erasure process was not addressed experimentally so much. Moreover, as pointed out by the authors, an experimental verification of the thermodynamic reversibility of the logically irreversible logic gate operation, such as the OR gate operation, is still missing. In the present paper, the authors introduced knowledges of the digital circuit and considered combinatorial logic gates. They realized a combinatorial logic gate by using a Si_{3N_4} micro-electromechanical cantilever. The output signal is a tiny deflection of the cantilever. To read the output signal, they performed precise deflection measurements by using a laser optical lever system. From the measurement of time-dependent displacement, they demonstrated that the energy dissipation is much smaller than $k_B T$ when the gate operation is performed slowly.

Although the idea of the authors is very interesting and their experiments would be sophisticated and reliable, to my impression, it is not so reasonable to regard a micro electromechanical cantilever as a combinatorial logic gate. As the authors explained in the paper, their setup perfectly works as a single combinatorial logic gate except for the readout process. The input signals are given by applying voltages to two probe electrodes in Fig. 2 (a). The authors can clearly distinguish 00,01,10 and 11 signals [Fig. 2 (c)]. They realized the OR gate (The same setup can be used as the AND gate) and the NOR gate. However, I think the output terminal is also a necessary constituent of a real logic gate. Usually, a combinatorial logic gate consists of relays or transistors as depicted in Fig. 1. The output signal is the voltage of one terminal V_{out} of the circuit in Fig. 1. One can connect the output terminal directly to an input terminal of another combinatorial logic gate. In this way one constructs, e.g. arithmetic circuits, adder, subtractor and comparator. However the setup proposed by the authors needs a laser optical lever system to read

the output signal. I naively imagine that this measurement apparatus could waste a lot of energy irreversibly. I wonder that this intermediate apparatus could spoil the idea to utilize micro electromechanical cantilevers as zero-power combinatorial logic gates.

Of course in the previous experiments [Refs. 6,7,8,9], the measurement apparatus would also waste energy. I think even a single-electron transistor electrometer used in Ref. 7 cannot avoid a lot of irreversible heat dissipation. However, in the feedback control experiments [6,7] as well as in the memory erasure experiments [8,9], the measurement apparatus can be well separated from the system if the back-action of the measurement is small. The role of measurement apparatus in the previous experiments is substantially different from that in the readout process in the author's combinatorial logic gate operation.

In order to strengthen their claim, I can suggest the authors to describe a method to transfer the output signal of one cantilever logic gate to the input terminal of another cantilever logic gate in a thermodynamically reversible manner. I think if the authors explain how to construct simple logic circuits, e.g. arithmetic circuits, potential readers will be convinced that micro electromechanical cantilevers actually work as combinatorial logic gates.

I think the direction of the author's research is of broad interest. I think the author's experiment is timely in the community of information thermodynamics and would stimulate further developments in this direction. I think the presentation of the paper is clear and the previous works are cited appropriately. However, I feel further analysis, which I mentioned above, would be desired.

Reviewer1:

We have revised the sentence that Reviewer 1 correctly pointed out.

Reviewer2:

We addressed all the points raised by the reviewer (please see the detailed answer below).

In our opinion, the reviewer mixed-up two aspects: the energy cost of the coupling (that is relevant for our claim) and the energy cost of the read-out (that is not relevant). We addressed the first one by showing new experimental data and critically discussed the second one.

Clearly the readout is a problem of practical relevance in any computer but is not pertinent to the computation process itself: in fact nobody can deny that a computer is computing even if there is nobody there to read the output.

Reviewer3:

The main point raised by this reviewer is partially coincident with one of the observations made by Reviewer 2 and is very relevant. We took seriously this point and acted in two ways:

1) We addressed the NOR gate presented in fig. 4 of our ms and measured the total amount of energy dissipated during the operation of this gate. In this case, as correctly pointed out by the reviewer, we have to deal with the transfer of information between one cantilever and the other. This is approximately $1.3e-21 \text{ J} = 0.31 k_B T$ and thus, our claim that the computation can be operated well below $k_B T$ still holds.

2) We have conceived and simulated the functioning of a full adder made by networking a set of 4 cantilevers. This shows that different cantilevers can be coupled in order to perform complex calculations. This is obtained without reading out the logic status of a single cantilever and feeding this to another one.

We hope that these arguments, supported by further evidence can contribute to clarify the relevant issue raised by the reviewer.

In view of the introduction of the new experimental data and the simulation results on the full adder, we believe that we have fully addressed all the relevant comments made by the reviewers.

We hope that this new improved version might meet the high standard criteria for publication on Nature Communications.

In the following, please, find, in red, a detailed answer to all the reviewers' criticisms.

Best regards,

Miquel López Suárez

Reviewer #1

The reviewer wrote:

I believe the authors have seriously considered my comments and made appropriate changes. I believe now the paper is ready for publication in Nature Communications as it is timely and important in the field thermodynamics; as I described in my previous report.

One minor point: In the introduction the authors write "This lack of linkage between logical and physical irreversibility". If I understand this correctly, the sentence should allude to a preceding discussion of the difference between logical and physical irreversibility. But as the words "physical irreversibility" are not mentioned until this point, the sentence makes little to no sense. I like the sentiment, but as written it is problematic.

Our answer:

We thank the reviewer for pointing this out. We modified the sentence as follows: "The claimed linkage[1] between logical and physical reversibility has animated a long debate[2] and, notwithstanding a recent clarification from a purely theoretical point of view[3], it still misses experimental verification."

Reviewer #2

The reviewer wrote:

I would like to thank the authors to revise the manuscript following the reviewer's comments. The manuscript was improved and it is easier to understand the concept that authors argue. I now understand that the point raised by the authors is that the total "dissipated" energy per cycle within the mechanical system can be smaller than kT . It was my fault that I did not fully understand this point. However, it is now rather clearer for me what is missing in the discussion and I believe that the manuscript does not include enough importance to warrant the publication in Nature Communications.

The energy cost for calculation should be evaluated between the initial configuration and the final configuration including all the related physical systems. In the final configuration, the calculation result should be readable otherwise the result cannot be utilized after the calculation.

Our answer:

We agree. In fact the result is readable by measuring the position of the cantilever tip.

The reviewer wrote:

Therefore the information should be somewhere stored even in their-called "combinational" logic.

Also in the present transistor-based logic system, the final results should be extracted from outside to get the results of calculation. Generally the dissipated energy of transistor-based devices might be evaluated only within the focused device itself but I believe that this is because the energy is much larger than kT and the energy used in the followed device to receive the transferred binary information can be neglected.

My largest question is; what is the reasonable definition of the energy required for computation. It is of course possible to define it as the energy spent only within the focused "combinational" device itself as authors did in this manuscript. However, if it is the case, the argument that "the operation energy can be smaller than kT " is confusing and misleading. The calculation result can be utilized only when the information was transferred to the next device and the energy consumed there should be discussed as the part of a whole calculation process. In the present setup, the position of cantilever read by the laser system changes some physical state there and an additional energy larger than $kT \log 2$ will be dissipated when the information is erased. I noticed that the authors already recognized this point as described in page 6 of revised manuscript but their discussion is still based on the energy dissipated only in the mechanical system.

Our answer:

The reviewer is invoking here two different aspects. One aspect is the "transfer" of the logic state to the next stage in the calculation. This is clearly relevant for the calculation of the energy dissipated and is the equivalent of the connection between two subsequent transistors. We agree that this is a relevant information that was not clearly discussed in the previous version of the ms. In this revised version we have explicitly measured the cost of coupling different cantilevers. As an example we have considered the NOR gate in fig.4. We have measured the energetic cost of operating this gate. It clearly includes the cost of the "transfer" of the logic state to the next stage. The result is $1.3e-21 \text{ J} = 0.31 k_B T$ and thus, our claim that the computation can be operated well below $k_B T$ still holds.

The second aspect invoked by the reviewer is the need for reading the output (the result of the computation). This operation is clearly external to the computing process itself and can or cannot need additional energy dissipation depending on the reading technique. Moreover, even if one wants to include this cost on the computation energetics, it has clearly to be diluted according to the number of computations performed. In principle one can operate a computers for billions of computations and never read the result until the end. At this point the impact of the reading cost on the single operation can be practically negligible.

The reviewer wrote:

The second question is on the sequential operation, raised from the application viewpoint, i.e. the fan-out. The energy dissipation might be lower than kT within the single device, but I am not sure if the output is

readable in the followed similar device. The optical method has enough high position sensitivity to distinguish the two states but transferring the information to next cantilever requires similarly high position sensitivity. I believe that some buffer device is required to amplify the signal and the energy spent therein should be considered.

Our answer:

This is a sensible point and, as anticipated above, we measured the energy dissipation associated with the connection of two cantilevers. This is $1.3e-21 \text{ J} = 0.31 k_B T$ and thus, our claim that the computation can be operated well below $k_B T$ still holds. Moreover we have also conceived and simulated a “full adder” combinational device by networking 4 cantilevers (see highlighted text on ms and additional information). The results show that these electromechanical devices can be coupled to perform more complicated operations giving a readable output. The simulation, performed considering the model presented in the article, considers no amplification or buffering device, nor other mechanisms.

The reviewer wrote:

In summary, I now agree with the argument given by the authors in this manuscript that the "dissipated" energy per cycle within the focused mechanical device can be smaller than kT . However, I believe that the experiment is straightforward and the statement, "the device can be operated with energy lower than kT " is true but highly misleading because the energy dissipated in the input and output system, as well as the amplifier required for the sequential operation was not taken into account.

Our answer:

The energy dissipated during the computation has been measured in our experiment and this is all the energy involved in the process. This includes both the mechanical energy dissipated for internal friction in the material and the electrostatic energy associated with the operation of the two electrical probes. There is no input and output energy change associated with the computation that is not taken into account.

Other energies that are not pertinent with the computation have not been considered, as the energy necessary to operate the electrostatic generator or the energy needed by the laser for the read-out of the final results. This energy might be important but are not pertinent to what is the paper claim: a logically irreversible computation can be operated with arbitrarily low energy dissipation, contrary to the claim made in [1].

The reviewer wrote:

Therefore, I cannot recommend the publication of this manuscript. A nice demonstration was given by for example ref. 8, where the energy required to erase the stored information was also taken into account in order to prove the asymptotic approach to the Landauer's limit.

Our Answer:

As we pointed out explicitly in our ms: “If we want to remember the final state we need to couple this device to a sequential device where a Landauer reset[13] might be required and a minimum dissipation of $k_B T \log 2$ needed.”. The difference of ref.[8] with our claim is that we are not storing or “erasing” information. We are doing a calculation and this can clearly be made without storing information (that require the use of sequential devices) by using purely combinational devices. Once again we stress that if, at the end of any computation, one wants to store information (and thus do a Landauer reset, provided that the state of the sequential device is not known) then one need to spend a given minimum amount. But also in this case such an amount will impact on the single computations for a quantity that has to be divided for the number of intermediate steps and thus might be (in practice) negligible.

Reviewer #3

The reviewer wrote:

I have read the paper "Sub $k_B T$ micro electromechanical irreversible logic gate" by Lopez-Suarez, Neri and Gammaitoni. The paper deals with a very interesting topic, the information thermodynamics, which attracts much attention. Recently, several experiments relevant for this topic have appeared as cited in the paper [Refs. 6,7,8,9]. In the paper, the authors stressed the difference between the logical irreversibility and the thermodynamic irreversibility. This is an important issue and has been theoretically discussed for a long time. Recently it has been a major interest in the community because of advances in work and heat measurement techniques of small systems.

The previous experiments [Refs. 6,7,8,9] have been more or less related to the Landauer's erasure principle. The memory erasure process is logically irreversible, but can be thermodynamically reversible. Although the minimum energy cost $k_B \log 2$ has been verified experimentally [Refs. 8 and 9], thermodynamic reversibility of the erasure process was not addressed experimentally so much. Moreover, as pointed out by the authors, an experimental verification of the thermodynamic reversibility of the logically irreversible logic gate operation, such as the OR gate operation, is still missing. In the present paper, the authors introduced knowledges of the digital circuit and considered combinatorial logic gates. They realized a combinatorial logic gate by using a Si_3N_4 micro-electromechanical cantilever. The output signal is a tiny deflection of the cantilever. To read the output signal, they performed precise deflection measurements by using a laser optical lever system. From the measurement of time-dependent displacement, they demonstrated that the energy dissipation is much smaller than $k_B T$ when the gate operation is performed slowly.

Although the idea of the authors is very interesting and their experiments would be sophisticated and reliable, to my impression, it is not so reasonable to regard a micro electromechanical cantilever as a combinatorial logic gate. As the authors explained in the paper, their setup perfectly works as a single combinatorial logic gate except for the readout process. The input signals are given by applying voltages to two probe electrodes in Fig. 2 (a). The authors can clearly distinguish 00,01,10 and 11 signals [Fig. 2 (c)]. They realized the OR gate (The same setup can be used as the AND gate) and the NOR gate. However, I think the output terminal is also a necessary constituent of a real logic gate. Usually, a combinatorial logic gate consists of relays or transistors as depicted in Fig. 1. The output signal is the voltage of one terminal V_{out} of the circuit in Fig. 1. One can connect the output terminal directly to an input terminal of another combinatorial logic gate. In this way one constructs, e.g. arithmetic circuits, adder, subtractor and comparator. However the setup proposed by the authors needs a laser optical lever system to read the output signal. I naively imagine that this measurement apparatus could waste a lot of energy irreversibly. I wonder

that this intermediate apparatus could spoil the idea to utilize micro electromechanical cantilevers as zero-power combinatorial logic gates.

Of course in the previous experiments [Refs. 6,7,8,9], the measurement apparatus would also waste energy. I think even a single-electron transistor electrometer used in Ref. 7 cannot avoid a lot of irreversible heat dissipation. However, in the feedback control experiments [6,7] as well as in the memory erasure experiments [8,9], the measurement apparatus can be well separated from the system if the back-action of the measurement is small. The role of measurement apparatus in the previous experiments is substantially different from that in the readout process in the author's combinatorial logic gate operation.

In order to strengthen their claim, I can suggest the authors to describe a method to transfer the output signal of one cantilever logic gate to the input terminal of another cantilever logic gate in a thermodynamically reversible manner. I think if the authors explain how to construct simple logic circuits, e.g. arithmetic circuits, potential readers will be convinced that micro electromechanical cantilevers actually work as combinatorial logic gates.

I think the direction of the author's research is of broad interest. I think the author's experiment is timely in the community of information thermodynamics and would stimulate further developments in this direction. I think the presentation of the paper is clear and the previous works are cited appropriately. However, I feel further analysis, which I mentioned above, would be desired.

Our answer:

We thank the reviewer for his/her comments. We valued very much the insights that were given and we acted in the direction suggested by the reviewer by doing the following:

1) We addressed the NOR gate presented in fig. 4 of our ms and measured the total amount of energy dissipated during the operation of this gate. In this case, as correctly pointed out by the reviewer, we have to deal with the transfer of information between one cantilever and the other. This is $1.3e-21 \text{ J} = 0.31 k_B T$ and thus, our claim that the computation can be operated well below $k_B T$ still holds.

2) We have conceived and simulated the functioning of a full adder made by networking a set of 4 cantilevers (see highlighted text on ms and additional information). This shows different cantilevers can be coupled in order to perform complex calculations, without the need for reading out the logic status of a single cantilever and feed this to another one. The simulation, performed considering the model presented in the article, considers no amplification or buffering device, nor other mechanisms.

We hope that this further experimental data and simulations can contribute to clarify the relevant issue raised by the reviewer.

Reviewer #2 (Remarks to the Author):

The authors clearly answered to my questions and added new experimental results concerning the energy cost to transfer the information between two mechanical systems. I now believe that the paper is nearly ready for the publication. My last comment is on the details of the energy estimation for the sequential operation in Fig.4. They provided only the final result $Q \sim 0.31kT$ but the experimental data (like Fig.3) and the way of calculation (i.e. which energies were taken into account, as shown in "Methods" section for a single cantilever) has not be provided. I recommend the author to show them at least in the Supplemental Materials.

Reviewer #3 (Remarks to the Author):

I have read the revised version of "Sub $k_B T$ micro electromechanical irreversible logic gate" by Lopez-Suarez, Neri and Gammaitoni. The authors considered my suggestions seriously. They addressed that the energy cost required to transfer the information between two cantilevers is as small as $0.3 k_B T$. If I understand their explanation correctly, this amount of energy is needed during a single NOR gate operation. Therefore, I wonder that it is not necessarily the energy cost required to transfer a digital signal between two different logic gates. They demonstrated a possible structure of the full-adder in Fig. S4. The circuit presented in Fig. S4 is different from what I naively anticipated: a network of combinatorial logic gates, such as NAND gates, whose input terminals and output terminals are connected each other. Anyway, Fig. S4 resolved my doubts. I understand that a cantilever logic gate does not have the input and the output terminal. The displacement of one cantilever itself becomes the input digital signal of another cantilever logic gate.

I am convinced that the information transfer between the cantilevers can be done without wasting energy. I trust that the authors would be able to construct zero-power arithmetic circuits. I appreciate the authors for their efforts to clarify my questions. I feel the paper is worth to be published, since as I wrote in my previous report, their results are timely in the community of information thermodynamics and would stimulate further developments.

Perugia, 19/05/2015

NATURE COMMUNICATIONS,

Referee response

Title: Sub $k_B T$ micro electromechanical irreversible logic gate

Authors: M. López-Suárez, I. Neri, L. Gammaitoni

We would like to thank both the referees for the useful comments and the constructive criticism, which led to an improvement of the manuscript. We have revised the manuscript to address their concerns.

Below we give a point-by-point response to the issues raised by the reviewers, marking them in red.

Reviewer #2:

The authors clearly answered to my questions and added new experimental results concerning the energy cost to transfer the information between two mechanical systems. I now believe that the paper is nearly ready for the publication. My last comment is on the details of the energy estimation for the sequential operation in Fig.4. They provided only the final result $Q \sim 0.31kT$ but the experimental data (like Fig.3) and the way of calculation (i.e. which energies were taken into account, as shown in "Methods" section for a single cantilever) has not be provided. I recommend the author to show them at least in the Supplemental Materials.

We added a section in the main text explaining the methodology to estimate the heat production limit in the case of two coupled cantilevers. We would like to thank again the referee for the useful comments and suggestions.

Reviewer #3:

I have read the revised version of "Sub $k_B T$ micro electromechanical irreversible logic gate" by Lopez-Suarez, Neri and Gammaitoni. The authors considered my suggestions seriously. They addressed that the energy cost required to transfer the information between two cantilevers is as small as $0.3 k_B T$. If I understand their explanation correctly, this amount of energy is needed during a single NOR gate operation. Therefore, I wonder that it is not necessarily the energy cost required to transfer a digital signal between two different logic gates. They demonstrated a possible structure of the full-adder in Fig. S4. The circuit presented in Fig. S4 is different from what I naively anticipated: a network of combinatorial logic gates, such as NAND gates, whose input terminals and output terminals are connected each other. Anyway, Fig. S4 resolved my doubts. I understand that a cantilever logic gate does not have the input and the output terminal. The displacement of one cantilever itself becomes the input digital signal of another cantilever logic gate.

I am convinced that the information transfer between the cantilevers can be done without wasting energy. I trust that the authors would be able to construct zero-power arithmetic circuits. I appreciate the authors for their efforts to clarify my questions. I feel the paper is worth to be published, since as I wrote in my previous report, their results are timely in the community of information thermodynamics and would stimulate further developments.

We would like to thank again the referee for the useful comments and suggestions.

Whit regards,

Dr. Miquel López-Suárez